# Selective Molecular Recognition of Low Density Lipoprotein Based on β-Cyclodextrin Coated Electrochemical Biosensor

**DOI:** 10.3390/bios11070216

**Published:** 2021-06-30

**Authors:** Huimin Wu, Fei Fang, Chengcheng Wang, Xiao Hong, Dajing Chen, Xiaojun Huang

**Affiliations:** 1MOE Key Laboratory of Macromolecular Synthesis and Functionalization, Department of Polymer Science and Engineering, Zhejiang University, Hangzhou 310027, China; wuhuimin1224@163.com (H.W.); 3090102659@zju.edu.cn (F.F.); 11629013@zju.edu.cn (X.H.); 2School of Medicine, Hangzhou Normal University, Hangzhou 311121, China; wangchengcheng@stu.hznu.edu.cn

**Keywords:** cholesterol, low-density lipoprotein, β-cyclodextrins, multiple interactions, molecular recognition, sensing

## Abstract

The excess of low-density lipoprotein (LDL) strongly promotes the accumulation of cholesterol on the arterial wall, which can easily lead to the atherosclerotic cardiovascular diseases (ACDs). It is a challenge on how to recognize and quantify the LDL with a simple and sensitive analytical technology. Herein, β-cyclodextrins (β-CDs), acting as molecular receptors, can bind with LDL to form stable inclusion complexes via the multiple interactions, including electrostatic, van der Waals forces, hydrogen bonding and hydrophobic interactions. With the combination of gold nanoparticles (Au NPs) and β-CDs, we developed an electrochemical sensor providing an excellent molecular recognition and sensing performance towards LDL detection. The LDL dynamic adsorption behavior on the surface of the β-CD-Au electrode was explored by electrochemical impedance spectroscopy (EIS), displaying that the electron-transfer resistance (Ret) values were proportional to the LDL (positively charged apolipoprotein B-100) concentrations. The β-CD-Au modified sensor exhibited a high selectivity and sensitivity (978 kΩ·µM^−1^) toward LDL, especially in ultra-low concentrations compared with the common interferers HDL and HSA. Due to its excellent molecular recognition performance, β-CD-Au can be used as a sensing material to monitor LDL in human blood for preventing ACDs in the future.

## 1. Introduction

Atherosclerotic cardiovascular disease (ACD), including coronary heart disease, cerebral infarction, and peripheral vascular disease, is one of the leading causes of death worldwide [1,2]. To date, many epidemiologic studies have shown that excess low-density lipoprotein (LDL) or oxidized modified low-density lipoprotein (OX-LDL) would promote the accumulation of cholesterol on the arterial wall, which can easily cause the occurrence of atherosclerosis [3,4]. LDL is a large protein composed of lipid and positively charged apolipoprotein B-100 (apoB100), which can be recognized by negatively charged adsorption materials via electrostatic interaction [5,6,7,8]. Fang et al. pointed out that in addition to the charge effect, saccharides or saccharide-like structures are also an important factor for LDL recognition [9]. Therefore, high LDL adsorption capacity contain charge effect and saccharides structures, such as heparin, chitosan derivatives, sulfated dextran, cucurbit, sodium alginate, polyacrylic acid, etc. [6,10]. Among them, β-cyclodextrins (β-CDs) are oligosaccharides consisting of seven glucose units, which present a toroidal form with a hydrophobic inner cavity and a hydrophilic outer surface [11,12,13,14,15]. β-CDs can act as molecular receptors owing to the multiple interactions, including electrostatic, van der Waals forces, hydrogen bonding, and hydrophobic interactions [16,17,18,19]. They can selectively interact with various guest molecules, such as small molecules, cationic or anionic guests, proteins, and polymer chains, to form stable inclusion complex or nanostructured supramolecular assemblies in their hydrophobic cavity, showing a high molecular recognition performance to LDL.

Currently, various analytical techniques are employed to assay protein levels, including high-performance liquid chromatography (HPLC) [20,21], enzyme-linked immunosorbent assay (ELISA) [22], fluorimetry [23], chromatographic methods [24], electrophoresis [25,26], mass spectroscopy [27], and chemiluminescence [28], which are typically expensive instruments with a long turnaround, requirement of sophisticated procedures, and well-trained operators. Thus, it is necessary to develop a sensitive and fast analytical method for LDL quantification to prevent and monitor atherosclerotic ACDs.

Electrochemical sensors based on β-CDs are an emerging diagnostic technology for rapid and accurate detection for target molecules [29,30,31,32,33,34,35]. For example, Abbaspour et al. designed an electrochemical sensor for simultaneous quantification of serotonin and dopamine using β-CDs/poly(N-acetylaniline)/carbon nanotube composite modified carbon paste electrode [30]. Xie et al. utilized host–guest interaction of β-CDs to eliminate the excessive small molecules, and combine the single-walled carbon nanotubes and β-CDs for excellent sensing performance for the detection of melamine [29]. Zhao et al. presented a three-in-one nanoplatform for self-assembly, cascade catalysis, and sensing, enabled by cyclodextrin modified gold nanoparticles (CD@AuNPs) [36]. Therefore, β-CDs are ideal absorption materials for LDL recognition and quantification via electrochemical method, which can convert the concentrations of LDL into measurable electrical signals timely and quickly.

In this study, a novel electrochemical sensing strategy for sensitively and selectively measuring LDL was designed based on the principle of multiple interactions between β-CDs and LDL. The electrochemical sensors for LDL recognition were fabricated by gold nanoparticles (Au NPs) electrodeposited electrode and SH-β-CD chemically self-assembled in Au modified electrode (Figure 1). The LDL was absorbed on the β-CD surfaces via multiple interactions and simultaneously converted the binding targets into electrochemical change of electrode interfaces. The LDL dynamic adsorption behavior on the surface of the β-CD-Au electrode was explored by electrochemical impedance spectroscopy (EIS), displaying that the electron-transfer resistance (Ret) values were proportional to the LDL (positively charged apolipoprotein B-100) concentrations from 2.5 to 25 μg·mL^−1^. The optimal adsorption time and concentration were also determined by impedance responses and faradaic impedance spectra. In addition, high selectively recognition of β-CD modified surface electrode was investigated by comparing with other single and complex proteins environment, including high-density lipoprotein (HDL) and human serum albumin (HSA).

## 2. Materials and Methods

### 2.1. Chemicals and Reagents

Low density lipoprotein (LDL, 98%), high-density lipoprotein (HDL, 98%), and human serum albumin (HSA) were purchased from Millipore (Massachusetts). Mono-(6-mercapto-6-deoxy)-b-cyclodextrin (SH-β-CD) was obtained from Zhiyuan Biotechnology Co., Ltd. (Shandong, China). Stainless steel substrates were purchased from Sigma-Aldrich (St. Louis, MO, USA). Milli-Q-water was prepared via ultrafiltration of distilled water to 18.2 MΩ·cm with an ELGA Classic UF system (Veolia Water Systems, France). All other reagents were supplied by Aladdin Reagent Co., Ltd. (Shanghai, China) and used as received without further purification, unless otherwise specified.

### 2.2. Apparatus

Morphological characterization of stainless-steel substrate, Au modified electrode, and β-CD-Au electrode were recorded using a field-emission scanning electron microscopy (FESEM, Hitachi S-4800, Japan) and atomic force microscope (AFM, SPI-3800N, Japan). The elemental content was obtained by X-ray photoelectron spectroscopy (XPS, PHI 5000c, PerkinElmer Instruments, Waltham, MA, USA). A three-electrode system was used in the experiment, including a working electrode, Ag/AgCl reference electrode, and Pt counter electrode. The applied potentials in all the measurements were vs. the Ag/AgCl reference electrode. The electrochemical impedance spectroscopy (EIS) was carried out with CHI660C electrochemical workstation (Chenhua, China) at 0.24 V, and its AC disturbance voltage was in the frequency range between 0.1 and 10^5^ Hz.

### 2.3. Fabrication of β-CD-Au Sensor for LDL Detection

#### 2.3.1. Preparation of Au Modified Electrode

As shown in Figure 2, a 0.2 mm diameter stainless steel needle was used as the support for the sensor. The stainless-steel needle is a promising electrode substrate for implantable sensors, which has been used in our group for continuous glucose testing [37]. The stainless-steel substrates were polished repeatedly with sandpaper (1500 mesh) and thoroughly cleaned by ultrasonication in distilled water and ethanol for 5 min. Next, we used constant potential method to electrodeposite Au NPs on needles. The electrodeposition coulomb was controlled at approximately 2 × 10^−2^ C. Next, the Au needles were washed with Milli-Q water and ethanol three times alternately. Finally, the Au modified electrodes were blown dry with high-purity argon.

#### 2.3.2. Preparation of β-CD Modified Au Electrode

The 1.0 mM SH-CD was dissolved in dimethylformamide (DMF) solution and bubbled nitrogen for about 10 min to remove dissolved oxygen. Next, the fabricated Au needles were immersed in above SH-CD solution at 25 °C in the dark for 24 h. Further, the Au needles were rinsed three times with DMF/ethanol solution (*v*/*v* = 1:1) followed by Milli-Q water to eliminate unbound thiol molecules. After rinsing cycles, the β-CD modified Au electrodes (β-CD-Au sensor) were dried in nitrogen and carefully preserved.

#### 2.3.3. Electrochemical Measurements of β-CD-Au Sensor

In order to investigate adsorption kinetics between LDL and β-CD-Au sensor, 20 μg/mL of LDL was dissolved in PBS solution (50 mM, pH 7.0) with different time (0, 5, 10, 15, 20, 25 min), containing 0.1 M KCl and 5 mM K_3_[Fe(CN)_6_]/K_4_[Fe(CN)_6_]. The fabricated β-CD-Au electrode and an Ag/AgCl electrode were used as the working electrode and reference electrode, respectively, for detecting different concentrations of LDL in the frequency range between 0.1 and 10^5^ Hz at 0.24 V.

In order to obtain optimal adsorption concentration between LDL and β-CD-Au sensors, different concentrations of LDL (0, 2.5, 5, 10, 15, 20, 25 μg/mL) was dispersed in PBS solution (50 mM, pH 7.0), containing 0.1 M KCl and 5 mM K_3_[Fe(CN)_6_]/K_4_[Fe(CN)_6_], respectively.

In order to verify the selectivity and anti-interference performance of fabricated β-CD-Au sensors, 20 μg/mL LDL and 20 μg/mL HDL was dissolved in PBS solution (50 mM, pH 7.0), containing 0.1 M KCl and 5 mM K_3_[Fe(CN)_6_]/K_4_[Fe(CN)_6_], respectively. In addition, different composition of mixed biproteins (HSA/LDL) were dissolved in PBS solution (50 mM, pH 7.0), containing 0.1 M KCl and 5 mM K_3_[Fe(CN)_6_]/K_4_[Fe(CN)_6_], respectively. All EIS measurements were carried out using β-CD-Au sensors at room temperature. Due to the possibility of LDL protein denaturation on electrodes during the test, the detector is used as a disposable sensor in this paper.

The limit of detection (LOD) was determined by the following Equation (1):LOD = 3.3 σ/S (1)
where σ is the standard deviation of the noise, while the S is the slope of linear calibration curve measured by the sensor.

## 3. Results and Discussion

### 3.1. Synthesis and Characterization of β-CD-Au Modified Electrode

The SH-β-CD was assembled on Au nanoparticles modified electrode, and the structure and morphology are shown in Figure 3. Compared with the Au modified electrode, the O, C, and S peaks appeared (Figure 3a), and their contents were about 40.50%, 19.20%, and 5.83% (Table 1) on the surface of β-CD-Au modified electrode, displaying β-CD assembled in Au modified electrode successfully via Au-S covalent bonding. Subsequently, morphologies of Au modified surfaces, α-CD-Au, β-CD-Au, and γ-CD-Au modified surfaces were characterized by AFM test. Compared to original Au modified surfaces, the CD-Au modified surfaces showed smaller roughness (Figure 3b). Generally, the self-assembled monolayers of molecules are arranged in an orderly and tight manner. The different sizes of CDs with mercapto groups lead to the different gap sizes, so that the roughness increased with the molecular size. The average roughness of α-CD-Au, β-CD-Au, and γ-CD-Au modified surfaces increased from 0.242 nm to 1.26, 1.33, 1.37 nm, respectively, which indicated a successful monolayer construction of CD on the Au modified surfaces.

The surface morphologies of bare electrode and Au modified electrode are shown in Figure 3c. The bare electrode needs to be polished with abrasive paper during the preparation, so the obvious scratches can be seen on the SEM images. Compared with the surface of bare electrode, Au nanoparticles were dispersed on the surface of β-CD-Au modified electrode.

### 3.2. Kinetics of LDL Adsorption on the β-CD-Au Modified Electrode

The impedance spectrum includes a semicircle portion corresponding to the electron-transfer-limiting process and a linear part resulting from the diffusion-limiting step of the electrochemical process [38]. The diameter of the semicircle exhibits the electron-transfer resistance (Ret) of the layer, which controls the electron-transfer kinetics of the redox probe at the electrode interface [39,40]. In Figure 4a, with the increase of the adsorption time, all diameters of the semicircles increased, and the increased amplitudes gradually became smaller in the time range of 15–20 min. There were many active sites on the surface of β-CD-Au modified electrode before 15 min, the adsorption rate was mainly affected by the diffusion rate, and the rate of adsorption was fast via monolayer absorption. In the range of 15–20 min, the adsorption rate was slower because the adsorption sites are gradually occupied. In the range of 20–25 min, the adsorption rate became faster again, probably due to the multilayer adsorption of LDL on the β-CD-Au modified electrode.

The impedance spectra are modeled using the equivalent circuit and the fitting analysis was examined by Zsimpwin software, which was depicted in Figure 4c. Comparing the actual data (red cubic) and fitting data (green cubic), the simulated data matched well with the original data, and the Chsq value was about 3.59 × 10^−4^. In this equivalent circuit (in the inset of Figure 4c), Rs represents the solution resistance, Ret represents the charge transfer resistance of the electrode interface, Zw represents the Warbug impedance, that is, the diffusion resistance of ions from the solution to the electrode surface, which reflects the characteristics of the mass transfer process, and Cdl represents the electric double layer capacitance [39]. Furthermore, Figure 4b depicted the relationship of Ret with incubation time. With the increase of the adsorption time, LDL was continuously adsorbed to the surface of the detector, leading to a linear increase of contact resistance in the first 20 min. However, the contact resistance increased rapidly after 20 min, because the LDL molecules formed a biofilm on the surface of the detector and resulting in the multilayer adsorption in the following step. The result was consistent with Figure 4a. The electrical impedance spectrum could reflect the performance of the surface electrode. In order to avoid the interference of the multilayer adsorption in this experiment, all subsequent experimental conditions were tested at 15 min.

### 3.3. LDL Adsorption on the β-CD-Au Modified Surface

The interaction of LDL with β-CD-Au modified electrode was investigated by EIS. Different concentrations of LDL were added in PBS solution and the corresponding Nyquist plots were shown in Figure 5a. The values of Ret linearly increased from 2.03 to 9.36 kΩ with the increase of LDL from 2.5 to 20 µg·mL^−1^ as reflected by the increase in diameter of the semicircle of the Nyquist plot. When the LDL level was higher than 20 µg·mL^−1^, the values of Ret was not linear and tended towards saturation. In detail, a linear relationship was extracted with a correlation efficiency of 0.988 in the range of 2.5–20 µg·mL^−1^ LDL concentrations, showing a sensitivity of 978 kΩ·µM^−1^ (Figure 5b). Thereafter, the greatly increased resistance value is owing to the multilayer adsorption of LDL on the β-CD-Au modified electrode. Furthermore, the negatively charged LDL at pH 7.0 also offers electrostatic force of repulsion to the electrons transferring from negative redox couple [Fe(CN)_6_]^3−^ and [Fe(CN)_6_]^4−^, thereby resulting in an increase in the Ret values. The result displayed that there was good adsorption between LDL and β-CD-Au modified electrode, which could be expected to be used in the detection of LDL in human blood in the future.

### 3.4. Selectivity of the β-CD-Au Modified Sensor

Selectivity is essential requirement for LDL sensor in POCT applications. There are various types of proteins in human blood. Among them, high-density lipoprotein (HDL) has a similar structure and surface chemistry to LDL but plays a opposite physiological role. It can transport excess cholesterol in the cells of the whole body back to the liver for decomposition, which is called “good cholesterol”. After the addition of HDL, the electrical impedance spectrums of β-CD-Au modified sensor remained unchanged (Figure 6a). However, when adding the same level of LDL in the same condition, the charge transfer resistance increased obviously (Figure 6b). Even though the size of HDL particles were (21.5 ± 6.5 nm) smaller than those of LDL (28.9 ± 9.2 nm), the adsorption capacity of β-CD to HDL was far less than that of LDL (Figure 6c), indicating excellent selective LDL adsorption for β-CD.

In addition, there are large amounts of human serum albumin (HSA) in human blood, with ellipsoid shapes of 36 nm in diameter. Therefore, we selected HSA-LDL proteins composites, to simulate and evaluate the selectivity of β-CD-Au modified needle sensor. As shown in Figure 7a,b, after adding HSA, the charge transfer resistance was unchanged, which was similar to HDL, because of the poor interaction between HAS and β-CD. With the increase of LDL levels in the mixed solution, the charge transfer resistance gradually increased, indicating high sensitivity and selectivity of this novel β-CD-Au modified sensor. In this experiment, the HSA level (1 mg·mL^−1^) was much higher than the concentration of LDL in the mixed protein solution. Furthermore, the LDL levels (5–20 µg·mL^−1^) in the present work were far less than the physiological levels (0.144–0.30 μM) in real human blood. The high level of HSA and trace concentrations of LDL suggested high sensitivity and promising application of β-CD-Au modified sensor in monitoring the LDL level in the future.

According to our previous work [9], the effect of hydrogen bond and the cavity of β-CD on the interaction between β-CD and LDL was particularly explored by surface plasmon resonance (SPR) analysis. The SPR results showed that such β-CD-modified surface exhibited good selectivity. The EIS data is consistent with the previous SPR test results of our group.

## 4. Conclusions

In conclusion, we reported the selective adsorption and sensing towards LDL based on β-CD via electrochemical impedance spectroscopy (EIS). The Au NPs were electrodeposited in bare electrodes and SH-β-CD was self-assembled in Au modified electrodes for LDL detection. LDL can be absorbed on the β-CD surfaces to form inclusion complexes via multiple interactions and simultaneously converted the LDL concentrations into measurable electrical signals. Moreover, the β-CD-Au modified surface could be used as a selective biosensor for LDL detection. The novel sensor performed high sensitivity (978 kΩ·µM^−1^) when operated in PBS buffer. Comparing with HDL and HSA, the β-CD-Au modified sensor exhibited a good selectivity toward LDL, especially in micro levels testing (2.5–20 µg·mL^−1^). The excellent specificity and sensitivity of β-CD opens up a new avenue to recognize and quantify LDL in human blood for preventing ACDs in the future.

## Figures and Tables

**Figure 1 biosensors-11-00216-f001:**
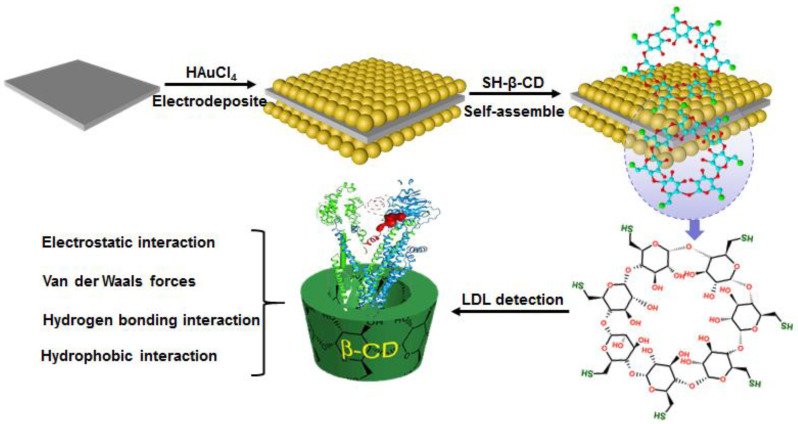
Schematic illustration of preparation of β-CD-Au modified needle for LDL recognition and quantification.

**Figure 2 biosensors-11-00216-f002:**
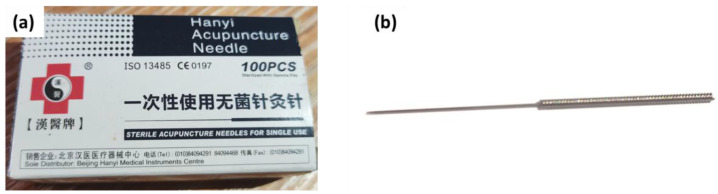
The images of (**a**) stainless-steel substrates by and (**b**) stainless-steel needle.

**Figure 3 biosensors-11-00216-f003:**
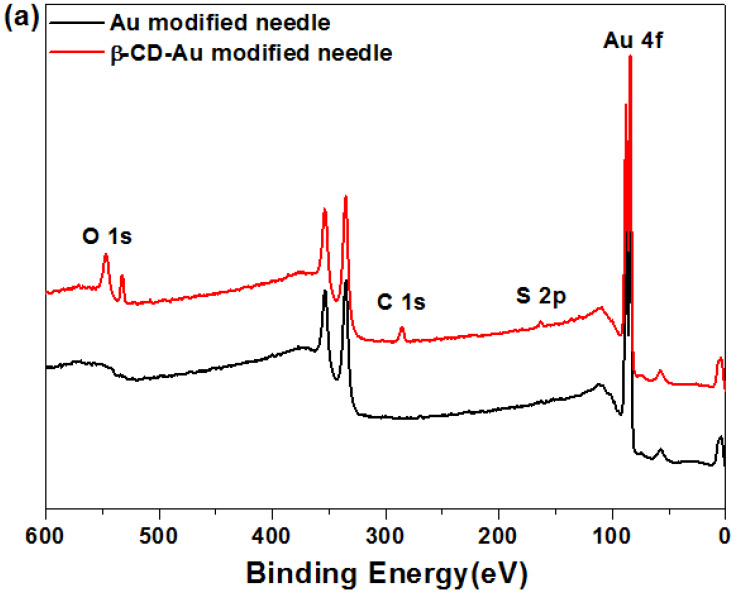
(**a**) XPS spectra for the surface of Au modified electrode and β-CD-Au modified electrode; (**b**) AFM images and roughness coefficients (Ra) of different modified surfaces; (**c****_1_**) SEM images for bare electrode, and (**c****_2_**) Au modified electrode.

**Figure 4 biosensors-11-00216-f004:**
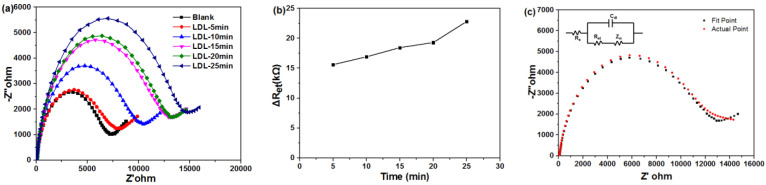
(**a**) Impedance responses with different time to the same concentration of LDL in PBS solution on the β-CD-Au modified electrodes. Faradaic impedance spectra were recorded in PBS (50 mM, pH 7.0) solution containing 0.1 M KCl and 5 mM K_3_[Fe(CN)_6_]/K_4_[Fe(CN)_6_]; (**b**) Relationship of ΔRet with the incubation time on the response of EIS; (**c**) Nyquist diagram for faradaic impedance spectra of LDL with 15 min in PBS (50 mM, pH 7.0) solution containing 0.1 M KCl and 5 mM K_3_[Fe(CN)_6_]/K_4_[Fe(CN)_6_]: actual (red cubic), fitting (green cubic), and the related Equivalent circuit for the impedance spectroscopy.

**Figure 5 biosensors-11-00216-f005:**
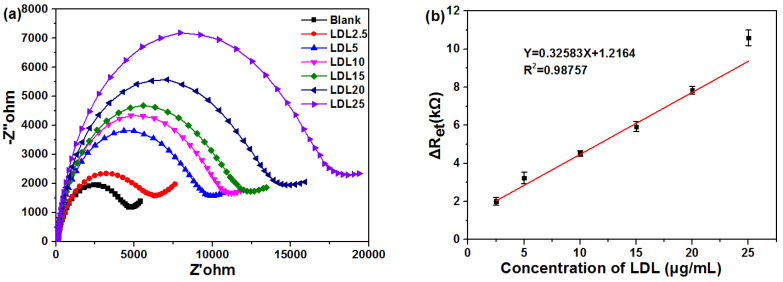
(**a**) Faradaic impedance spectra that corresponded to the fabricated sensor at 25 °C, before and after incubating with different concentrations of LDL in PBS (50 mM, pH 7.0) solution containing 0.1 M KCl and 5 mM K_3_[Fe(CN)_6_]/K_4_[Fe(CN)_6_]; (**b**) Calibration curve for the relationship of ΔRet with the LDL concentration.

**Figure 6 biosensors-11-00216-f006:**
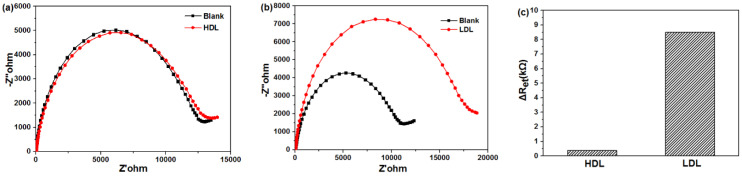
Faradaic impedance spectra that corresponded to the fabricated sensor at 25 °C, before and after incubating with same concentrations of (**a**) HDL, 20 μg/mL or (**b**) LDL, 20 μg/mL in PBS (50 mM, pH 7.0) solution containing 0.1 M KCl and 5 mM K_3_[Fe(CN)_6_]/K_4_[Fe(CN)_6_]; (**c**) ΔRet of different lipoprotein calculated to faradaic impedance spectra.

**Figure 7 biosensors-11-00216-f007:**
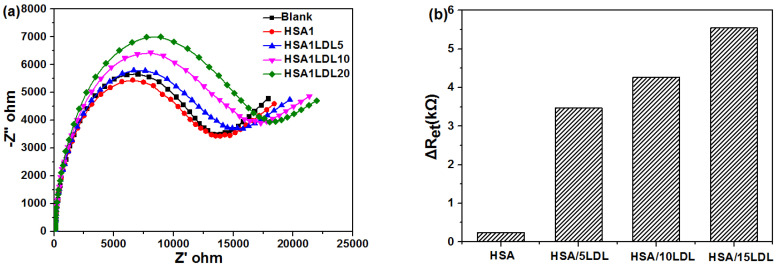
(**a**) Nyquist diagram of faradaic impedance spectra and (**b**) ΔRet of binary proteins for β-CD based sensor incubating in HSA/LDL binary proteins in PBS (50 mM, pH 7.0) solution containing 0.1 M KCl and 5 mM K_3_[Fe(CN)_6_]/K_4_[Fe(CN)_6_].

**Table 1 biosensors-11-00216-t001:** Elemental analysis on the surface of Au modified electrode and β-CD-Au modified electrode.

Samples	C 1s (%)	O 1s (%)	S 2p (%)	Au 4f (%)
Au modified electrode	0.82	0.32		98.86
β-CD-Au modified electrode	40.50	19.20	5.83	34.16

## Data Availability

Not applicable.

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
