# Peer review of "Selective Molecular Recognition of Low Density Lipoprotein Based on β-Cyclodextrin Coated Electrochemical Biosensor"

_biosensors, 2021, doi:10.3390/bios11070216_

Round 1
Reviewer 1 Report
In this study, the authors report a novel electrochemical sensing strategy for sensitively and selectively measuring LDL was designed based on the principle of multiple interactions between β-CDs and LDL. The paper is clear and interesting. Few minor comments need to be addressed before publication. Could the author estimate the LOD? How the error bars reported in Figure 4b have been estimated?
Reviewer 2 Report
Dear All
The present manuscript reported that Selective molecular recognition of low density lipoprotein based on β‑cyclodextrin coated electrochemical biosensor. The results can be interesting to bio-sensing analysis. It needs to improve and you should answer all comments carefully and do more experiments.
*Page 3/line 120; 2.3.2 Preparation of β-CD modified Au electrode/
Why did you put in the dark place??
**Page 3/ line 108/ You mentioned: “stainless steel substrates” , please add more information? Put an image from your electrode in the paper. please add dimensions? where did you buy or ….. it ? as a reviewer I do not understand, I guess the readers too.
*Page 3, line 111: “The Au NPs were electrodeposited on needles by cyclic voltammetry (CV) scanning at 111 the -2.5 V (vs Ag/AgCl) at 0.1s cycle interval.”
I do not understand?!! You used CV or constant potential method? you applied -2.5 V. it is wrong for CV, please modify it. In CV you scan potential in a potential range.
Please add CV of deposition process. I like to see the CV and peaks.
* Page 3/line 126/ LDL was dissolved in PBS solution (50 mM, pH 7.0)?
Please tell us, why did you use pH 7.0 ? What is the solubility of LDL in water and pH 7 ? please add references.
*page 8/line 247/” because of the poor interaction between HAS and β-CD. With the increase of LDL levels in the mixed solution, the charge 248 transfer resistance gradually increased, indicating high sensitivity and selectivity of this 249 novel β-CD-Au modified sensor”’.
It is not a scientific conclusion; it is a just observation. Please tell us, how the sensor showed higher selectivity to LDL? why LDL˃HDL. You mentioned both have the same structure or very similar. You have to prove the interaction and mechanism of sensing.
* please calculate the limit of detection?
* about the reputability of the sensor? how many times can we use one sensor?
* after the first measurement, how did you recover the surface of electrode for next measurement?
* conclusion section “selective adsorption and sensing”, but you did not have any scientific statement to show “selective adsorption”. Please add mechanism of interaction between BC and LDL. IR method can help?
* please tell us, why did you choose “Au NPs”? Explain and add related references.
I am waiting for extra explanations and new measurements (extra), then I read it again. Now, no chance for acceptance. So Major revision.
Round 2
Reviewer 2 Report
it has been improved well.
it can be accepted after M.R.
- Ref. 9 and 10, Journal title is wrong.: Langmuir : the ACS journal of surfaces and colloids 2018;34:8163-9. ??!!, please re-check all Ref.
- please add LOD to the paper, just in the response letter is not enough.
- please check the deposition method in the whole paper.
